# Association between Chronotype and Physical Behaviours in Adolescent Girls

**DOI:** 10.3390/children10050819

**Published:** 2023-04-30

**Authors:** Tatiana Plekhanova, Emily Crawley, Melanie J. Davies, Trish Gorely, Deirdre M. Harrington, Ekaterini Ioannidou, Kamlesh Khunti, Alex V. Rowlands, Lauren B. Sherar, Tom Yates, Charlotte L. Edwardson

**Affiliations:** 1Diabetes Research Centre, University of Leicester, Leicester LE5 4PW, UK; 2NIHR Leicester Biomedical Research Centre, Leicester LE5 4PW, UK; 3Department of Nursing and Midwifery, University of the Highlands and Islands, Inverness IV2 3JH, UK; 4School of Psychological Sciences and Health, University of Strathclyde, Glasgow G1 1XQ, UK; 5School of Sport, Exercise and Health Sciences, Loughborough University, Loughborough LE11 3TU, UK

**Keywords:** physical activity, sleep, sedentary behaviour, circadian preference

## Abstract

The aim of this study was to (1) describe accelerometer-assessed physical behaviours by chronotype, and (2) examine the association between chronotype and accelerometer-assessed physical behaviours in a cohort of adolescent girls. Chronotype (single question) and physical behaviours (GENEActiv accelerometer on the non-dominant wrist) were assessed in 965 adolescent girls (13.9 ± 0.8 years). Linear mixed-effects models examined the relationships among chronotype and physical behaviours (time in bed, total sleep time, sleep efficiency, sedentary time, overall, light and moderate-to-vigorous physical activity) on weekdays and weekend days. Over the 24 h day, participants spent 46% sedentary, 20% in light activity, 3% in moderate-to-vigorous physical activity, and 31% in ‘time in bed’. Seventy percent of participants identified as ‘evening’ chronotypes. Compared to evening chronotypes, morning chronotypes engaged in less sedentary time (10 min/day) and had higher overall physical activity (1.3 mg/day, ~30 min of slow walking) on weekdays. Most girls identified as evening chronotypes with a large proportion of their day spent sedentary and a small amount in physical activities which may be exacerbated in evening chronotypes on weekdays. The results maybe be important for programmes aiming to promote physical activity in adolescent girls.

## 1. Introduction

Chronotype is an individual’s preference to start their day either early or late and the time of day at which they feel their best [1]. Morning types wake up earlier and show performance peaks in the morning, whereas evening types wake up later and show a peak in performance in the evening. Chronotype is not fixed and has been shown to change throughout an individual’s lifespan and in different environments (e.g., the altitude/latitude of residence or exposure to light) [1]. During adolescence, young people typically shift towards an evening preference but are expected to rise early for school and social activities, e.g., sports [2]. This can lead to reduced sleep duration and a mismatch between biological clock and imposed school and social schedules [2].

Previous studies reported that evening chronotype in adolescents was associated with greater adiposity [3], depressive symptoms [4], health-impairing behaviours such as smoking and alcohol use [5], and worse school performance [4]. Additionally, evening chronotypes were shown to spend more time watching television and at the computer [6,7], less time engaging in physical activity [8,9,10], and, on weekdays, sleep less compared to morning types [7]. However, much of this evidence has been mostly based on self-reports of these behaviours. Device-based measures of physical behaviours, such as accelerometers, can capture the full 24 h day of physical behaviours including physical activity, sedentary behaviour, and sleep quantity and quality (e.g., sleep efficiency).

Because low physical activity, high sedentary time, and insufficient sleep (<8 h) in adolescents are associated with poor health outcomes [11] and worse school performance [12], exploring the differences in these potentially modifiable behaviours across chronotypes might help to inform future lifestyle interventions. Therefore, the aim of this study was (1) to describe accelerometer-assessed physical behaviours by chronotype and (2) to examine the association between chronotype and physical behaviours in a cohort of adolescent girls.

## 2. Materials and Methods

### 2.1. Study Design and Participants

This was a secondary data analysis that used cross-sectional data from the final data collection time-point (14-month follow-up) from a randomised controlled trial of a school-based physical activity intervention (Girls Active) [13]. In brief, the Girls Active intervention aimed to increase physical activity levels in adolescent girls by providing a support framework to schools to review and/or change their physical activity culture and practices. All state schools within Leicester, Leicestershire, and Rutland (LLR) (*n* = 56) and those schools that were geographically close to LLR (*n* = 26) with a Key Stage 3 (KS3: age 11–14 years) were eligible and were sent invitations to take part in the randomised controlled trial. Of these, 20 school principals provided consent for their school to take part between April to June 2016. All eligible participants aged 11–14 and in years 7, 8 and 9 were given an information pack with an opt-out consent form for their parents to complete. Of those that did not return an opt-out consent, 90 girls from each school were randomly selected using a random number generator to take part. A question on chronotype was included at the final data collection only (14-month follow-up). Participants who attended the final follow-up (*n* = 1361) were asked the chronotype question only after they first completed main testing procedures (*n* = 1298). Only those who had valid accelerometer data and answered the chronotype question were included in the analyses. Ethics approval was obtained from the University of Leicester ethics sub-committee for Medicine and Biological Sciences. The girls themselves provided verbal assent before any measurements took place.

### 2.2. Measures

Accelerometer variables: Participants were asked to wear the GENEActiv accelerometer (GENEActiv Original, Activinsights Ltd., Kimbolton, UK) 24 h per day for 7 days on the non-dominant wrist. A verbal explanation of the accelerometer was given to the participants in small groups and a demonstration was given on how to wear it. Participants were offered a gift voucher (£5) on return of the accelerometer with valid wear data. Accelerometers were configured to record at a frequency of 100 Hz. Data were downloaded using GENEActiv PC software vv.3.2 and accelerometer files were analysed with R-package GGIR v1.2–11 [14]. Signal processing in GGIR includes autocalibration using local gravity as a reference [15]; detection of non-wear; and calculation of the average magnitude of dynamic acceleration corrected for gravity (Euclidean Norm minus 1 g with negative values rounded up to zero, ENMO), averaged over 5 s epochs and expressed in milli-gravitational units (mg). Non-wear was imputed using the default setting; that is, invalid data were imputed by the average at similar time points on different days of the week [15]. Participants were excluded if post-calibration error was >0.01 g (10 mg), they had <3 days of valid wear (defined as >16 h per day) [16], or if wear data were not present for each 15-min period of the 24 h cycle. Variables of interest across weekdays and weekends were sedentary time (time accumulated during the waking day below 40 mg) [17], minutes of moderate-to-vigorous physical activity (MVPA; time accumulated with an acceleration >200 mg) [18], light activity (time accumulated with an acceleration between 40 and 200 mg) [18], average acceleration in mg (a proxy for overall physical activity), time in bed (hours, time between sleep onset and wake time), total sleep time (hours, time spent sleeping), and sleep efficiency (%, the ratio of total sleep time to time in bed). Sleep characteristics were derived using an automated sleep detection algorithm [19]. This algorithm facilitates detection of the sleep period time window (SPT-window) without the use of sleep diaries.

Chronotype: Chronotype was assessed using one item (item 19) from Morningness–Eveningness Questionnaire [20]. Participants were asked to choose one of four options: ‘’are you (1) definitely morning, (2) more morning than evening, (3) more evening than morning, or (4) definitely evening”. Chronotype data was collected only after main testing procedures were completed and at 14-month follow-up.

Covariates: Age and ethnicity (categorised into white European or non-white European) were self-reported by participants. Height (portable stadiometer) and weight (Tanita SC330S bioimpedance scale) were measured to the nearest 0.1 cm and 0.1 kg, respectively. Body mass index (BMI) was calculated as weight/height^2^. BMI z-score was calculated using BMI and age to provide a standardised measure relevant to the UK population [21].

### 2.3. Statistical Analyses

Since the Girls Active intervention had no effect on the primary outcome (moderate-to-vigorous activity) at 14-month follow-up [22], control and intervention groups were combined for the analysis. Participants were included in the analysis if they answered the chronotype question and had at least 2 weekdays and 2 weekend days of valid accelerometer data (defined as >16 h/day). “Definitely morning” and “more morning than evening” were grouped into the “morning” category, and “definitely evening” and “more evening than morning” were grouped into the “evening” category. The proportion of the girls meeting the guidelines for MVPA (at least 60 per day) [23] and total sleep time (8–10 h per night for adolescents aged 13–17) [24] was also reported across all days, weekdays and weekends. The proportions were calculated based on average daily MVPA (i.e., if >60 min/day) and total sleep time (i.e., if between 8–10 h/night) across valid days. Because these were similar across chronotypes, the proportions are reported for the whole sample only. The variables were tested for normality (using Q–Q plots). Linear mixed-effects models examined the relationships between chronotype and physical behaviours (time in bed, total sleep time, sleep efficiency, overall activity, sedentary time, light activity, and MVPA). All linear models were fitted separately for weekdays and weekends. For sleep variables, weekend nights were defined from Friday to Saturday and weekday nights as Sunday–Thursday. Models accounted for school-level clustering, age, ethnicity, BMI z-score, multiple deprivation index (IMD), and randomisation group (intervention or control). The fit of the data was assessed by inspecting residuals vs. fitted values (without transformations of the variables that were not normally distributed). These were found to be randomly scattered and indicated a good fit. R Project for Statistical Computing (v. 3.6.1) was used. Alpha was set at 0.05.

## 3. Results

Of the 1361 participants who attended the 14-month follow-up, 965 (71%) provided valid data for the analyses (mean age (SD) 13.9 ± 0.8 years, 73% white European, and 30% overweight or obese) (Figure 1).

Most girls (70%) identified themselves as an ‘evening’ chronotype. Participants did an average of 42.0 (19.7) minutes per day of MVPA. Table 1 presents the characteristics of the participants.

The 24 h profile of physical behaviours split by weekdays and weekends is presented for the whole sample and by chronotype category in Figure 2. In the whole sample and across chronotypes, participants spent a slightly higher proportion of the day in light-intensity physical activity and MVPA and less time sedentary on weekdays compared to weekends (Figure 2a). Morning and evening chronotypes displayed a similar 24 h profile of physical behaviours (Figure 2b,c).

The proportion of the girls meeting the guidelines for MVPA and sleep is presented in Table 2. Overall, 16% of the participants met the guidelines for MVPA and 22% met the guidelines for total sleep time. The number of participants meeting MVPA guidelines was similar on weekdays and weekends, but more girls achieved the recommended hours for total sleep time on weekends (40%) compared to weekdays (20%).

When examining the association between chronotype and physical behaviours (Table 3), identifying as a morning chronotype was associated with 10 min/day less spent in sedentary time (β = −0.17, 95% CI: −0.33, −0.01, *p* = 0.041) and higher overall physical activity on weekdays compared to an evening chronotype (β = 1.28, 95% CI: 0.16, 2.41, *p* = 0.025). No other associations between chronotype and any other physical behaviours were found.

## 4. Discussion

This study aimed to describe 24 h accelerometer-assessed physical behaviours by chronotype in a large sample of adolescent girls and examine the association between these physical behaviours and chronotype. The findings from the present study demonstrate that most adolescent girls in this sample had a preference for eveningness. Both morning and evening chronotypes spent a large proportion of their 24 h day sedentary and only a small amount in physical activities on both weekdays and weekends. This pattern was similar across chronotypes. Compared to morning chronotypes, evening chronotypes engaged in more sedentary time (10 min per day) and had lower overall physical activity (i.e., average acceleration) on weekdays, with the 1.3 mg/day difference per day approximating 30 min of slow walking [25].

In the present study, approximately 70% of the girls were classified as evening chronotypes. Given that the evening chronotype is associated with poorer academic achievement [4], and worse mental [4] and physical health [3], interventions could target circadian factors. There is some evidence showing that transdiagnostic sleep and circadian intervention can reduce eveningness in evening-type adolescents [26] and that a combination of cognitive behavioural therapy with light therapy can improve several sleep dimensions (e.g., earlier sleep onset times and longer total sleep time) and daytime functioning (e.g., reduced daytime sleepiness) in adolescents with delayed sleep phase disorder [27].

In the present study, participants spent the majority of their time in sedentary activities (46%), one-third (31%) in ‘time in bed’, and only a small proportion of the day in light activity (20%) and moderate-to-vigorous activity (3%). The proportion spent in each behaviour was similar across weekdays and weekends and across chronotypes. Very similar proportions for the 24 h activity profile assessed using accelerometers were found in a sample of 119 adolescent females across New Zealand [28]. However, it should be highlighted, that the participants in the sample from New Zealand were slightly older (16.8 vs. 13.9 years) and different accelerometer placement (worn on the waist) and processing methods were used. Studies that used wrist-worn accelerometers and processing methods such as in the present study have reported comparable times spent in different activities in girls of a similar age. For instance, both Sanders et al. [29] and Fairclough et al. [30] reported approximately 46 min per day of moderate-to-vigorous physical activity. Fairclough et al. [30] also reported similar durations for sleep (7 h), light activity (~270 min), and sedentary behaviours (~620 min) compared to the findings from the present study. Current 24 h movement guidelines for children and adolescents aged 5–17 recommend at least 60 min of moderate-to-vigorous physical activity, less than two hours of sedentary screen time, and 8–10 h of sleep for adolescents aged 13–18 each day [23,24]. In the present study, only 16% and 22% met the recommended guidelines for moderate-to-vigorous physical activity and sleep, respectively. Therefore, it is important to understand the 24 h activity profile for future interventions.

Previous studies have demonstrated an association between evening chronotype and lower physical activity and higher sedentary time, assessed with self-report, compared to morning types [6,8,10]. These findings were corroborated by Merikanto et al., who reported that eveningness was associated with lower device-measured overall, light, and moderate-to-vigorous physical activity and higher sedentary time [9]. In the present study, evening types had lower overall activity and spent more time sedentary on weekdays. It has been suggested that evening types have lower physical activity because they struggle to find a suitable time in line with their preference to be active [31]. Additionally, evening chronotypes experience more fatigue and perform worse physically when physical activities take place in the first part of the day rather than in the evening [31]. It is also possible that evening chronotypes have less energy due to sleep difficulties [32]. Previous studies have reported that evening chronotypes have more sleep difficulties such as insomnia symptoms and daytime sleepiness than morning chronotypes [33]. These difficulties may result from the mismatch between early school times and a preference towards eveningness. This mismatch may then lead to accumulating more sleep debt during school days, creating social jetlag [33]. In turn, insufficient sleep duration in adolescents has been shown to be associated with lower physical activity levels [34]. Given the significance of physical activity for adolescent health, it is important to understand the impact of chronotype on such health behaviours to appropriately target interventions. Interventions could include scheduling physical activities at a time suitable for adolescents with an evening chronotype to better motivate them for physical activities [9]. Timing of physical activity may help evening chronotypes in keeping active as well as ensuring that they obtain sufficient sleep to prevent daytime tiredness [9].

The differences in results between the present study and that of previous research could be due to differences in the assessment of physical behaviours and chronotype. Physical activity and sleep measured with self-report are subject to response and memory biases [35]. Furthermore, a range of questionnaires are used across the research to assess chronotype, and studies which utilised device-based measures were based on activity counts, not raw data, thus limiting comparability between studies.

### Strengths and Limitations

Strengths of our study include the large multi-ethnic population and the accelerometer-assessed physical behaviours. However, the analysis herein was based on data collected opportunistically as part of the final follow-up measurement in a larger cluster randomised controlled trial. As a result, the associations of chronotype with important outcomes such as academic achievement or mental or physical health could not be examined. The chronotype question was determined from a single question rather than a full questionnaire in order to reduce participant burden at the end of the study; however, Merikanto et al. demonstrated that item 19 of the MEQ was highly correlated with the full questionnaire [9]. Additionally, our population characteristics were limited by only including one gender (female) and girls from one geographical location (the East Midlands, UK).

## 5. Conclusions

Evening chronotype was highly prevalent in this sample of adolescent girls. Regardless of chronotype, the girls spent approximately half of their 24 h day sedentary and only a small proportion of the day in physical activities. The findings of this study provide valuable evidence of the chronotype of adolescent females and relationships with two lifestyle behaviours and suggest that adolescents with a preference for eveningness may be a particular target for intervention.

## Figures and Tables

**Figure 1 children-10-00819-f001:**
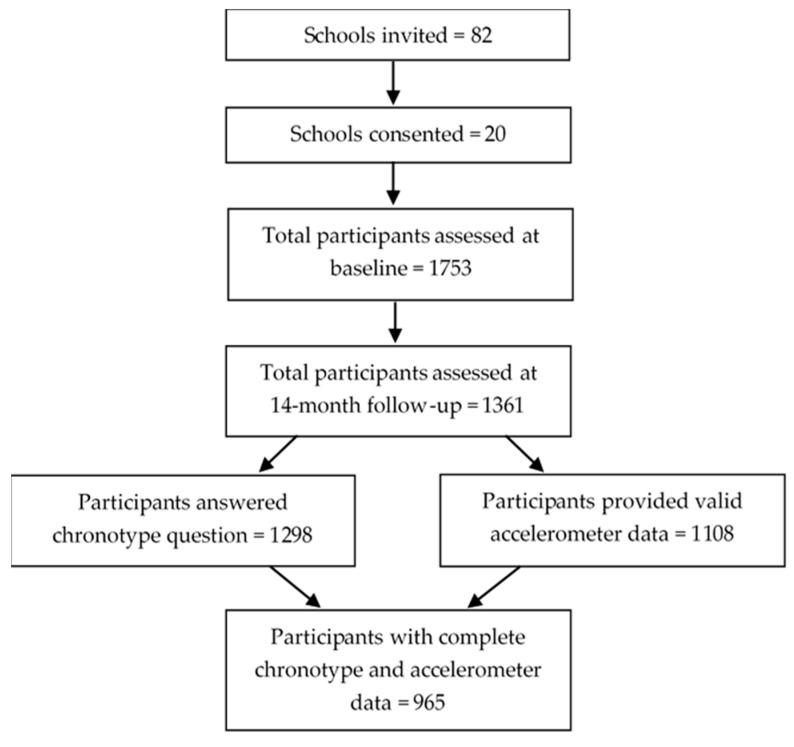
Participant flow chart.

**Figure 2 children-10-00819-f002:**
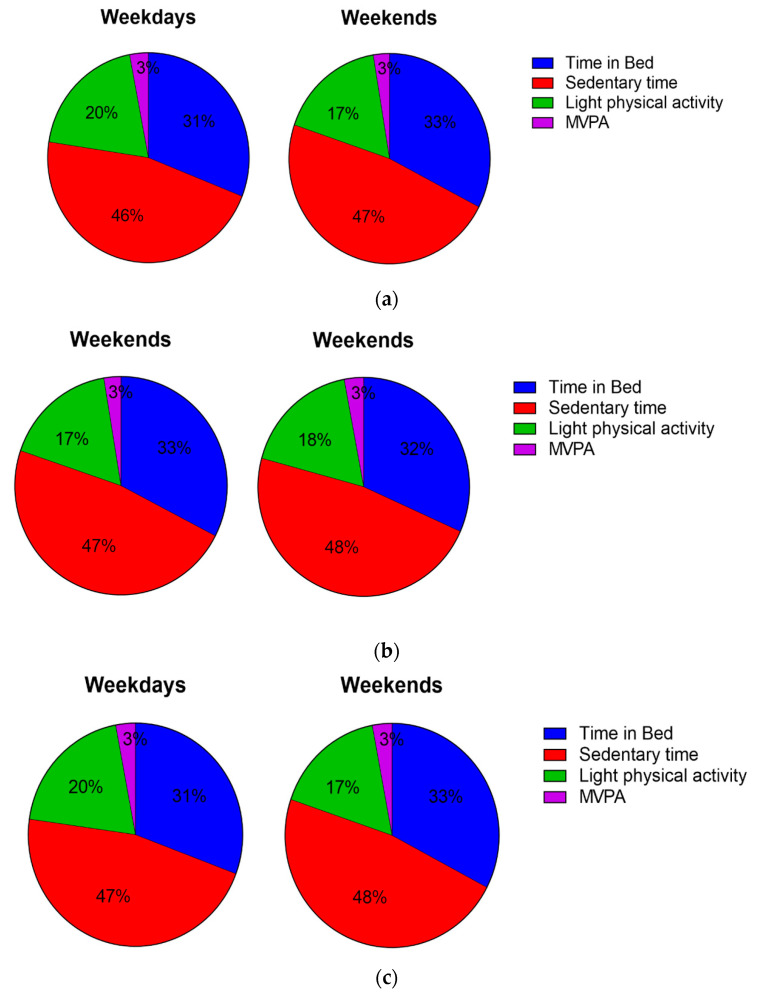
(**a**) 24 h physical behaviour profile for weekdays and weekends across the whole sample. Percentage of the day spent in each of the physical behaviours. (**b**) 24 h physical behaviour profile for weekdays and weekends for morning chronotype. Percentage of the day spent in each of the physical behaviours. (**c**) 24 h physical behaviour profile for weekdays and weekends for evening chronotypes. Percentage of the day spent in each of the physical behaviours.

**Table 1 children-10-00819-t001:** Characteristics of the whole sample and stratified by chronotype.

Characteristic	Whole Sample (*N* = 965)	Morning Chronotype (*N* = 291)	Evening Chronotype (*N* = 674)
	Mean (SD) or *n* (%)		
Age, years	13.9 (0.8)	13.8 (0.8)	14.0 (0.8)
Ethnicity categories White European Non-white European	703 (73%) 260 (27%)	204 (71%) 85 (29%)	499 (74%) 175 (26%)
BMI, z-score	0.37 (1.3)	0.32 (1.3)	0.39 (1.3)
BMI category Underweight Normal weight Overweight Obese	25 (2%) 637 (66%) 194 (20%) 94 (10%)	15 (5%) 197 (69%) 48 (17%) 25 (9%)	28 (4%) 456 (68%) 123 (18%) 59 (9%)
IMD rank score	16,814 (9267)	16,155 (9313)	17,095 (9240)
IMD decile score ^a^	5.6 (2.8)	5.4 (2.9)	5.7 (2.8)
Accelerometer variables MVPA, min/day Light PA, min/day Sedentary, min/day Overall daily PA, mg Total sleep time, h/day Time in bed duration, h/day Sleep efficiency, %/night Average wear days, valid days	42.0 (19.7) 271.8 (47.5) 672.7 (67.4) 34.7 (8.3) 7.6 (0.6) 8.6 (0.7) 81.4 (5.6) 6.9 (0.4)	42.9 (18.7) 276.3 (45.6) 666.1 (66.0) 35.6 (8.2) 7.6 (0.6) 9.4 (0.7) 80.4 (5.8) 6.9 (0.4)	41.3 (19.9) 269.8 (48.1) 678.3 (66.2) 34.3 (8.3) 7.6 (0.7) 9.3 (0.7) 81.3 (5.5) 6.9 (0.5)

BMI = body mass index; IMD = multiple deprivation index; mg = milligravitational units; MVPA = moderate-to-vigorous physical activity; PA = physical activity; SD = standard deviation. ^a^ IMD decile scores range from 1 to 10: 1 is the least deprived and 10 is the most deprived.

**Table 2 children-10-00819-t002:** Number and proportion of participants meeting the sleep and physical activity guidelines.

	N (%) Meeting MVPA Guidelines (At Least 60 min/day)	N (%) Meeting Sleep Guidelines (8–10 h/night)
Whole week	156 (16.2%)	212 (22%)
Weekdays	182 (18.9%)	190 (19.7%)
Weekends	168 (17.4%)	390 (40.4%)

MVPA = moderate-to-vigorous physical activity.

**Table 3 children-10-00819-t003:** Mixed-effects models for the association between chronotype and physical behaviours by weekdays and weekends (reference = evening chronotype).

Exposure:	Chronotype (Weekdays)		Chronotype (Weekends)	
Outcome:	b (95% CI)	*p*	b (95% CI)	*p*
Total sleep time (h)	0.06 (−0.44, 0.17)	0.254	−0.13 (−0.26, 0.30)	0.056
Time in bed (h)	0.05 (−0.07, 0.17)	0.436	−0.02 (−0.17, 0.13)	0.774
Sleep efficiency (%)	0.01 (−0.07, 0.01)	0.651	−0.01 (−0.02, 0.01)	0.060
Sedentary time (min)	−0.17 (−0.33, −0.01)	0.041	0.03 (−0.19, 0.25)	0.771
MVPA (min)	1.77 (−0.98, 4.53)	0.206	−1.06 (−4.57, 2.45)	0.553
Overall activity (mg)	1.28 (0.16, 2.41)	0.025	0.44 (−1.13, 2.01)	0.583
Light activity (min)	4.00 (−2.69, 10.61)	0.243	6.40 (−3.51, 16.31)	0.205

Data displayed as beta-coefficients (95% CI). Adjusted for school-level clustering, age, ethnicity, BMI z-score, multiple deprivation index rank score, and randomisation group. MVPA = moderate-to-vigorous physical activity.

## Data Availability

Data are available upon reasonable request.

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
