# Peer review of "Association between Chronotype and Physical Behaviours in Adolescent Girls"

_children, 2023, doi:10.3390/children10050819_

Round 1
Reviewer 1 Report
The paper presents an empirical study investigating the relationship between chronotype and physical behaviors in adolescent girls.
The research topic is of relevance for both research and practice.
The manuscript is well written and the study seems properly conducted.
Important strengths include the large sample size of almost 1000 participants and the detailed physical activity assessments based on accelerometers.
Overall, participants spent 46% sedentary, 20% in light and 3% in
moderate-to-vigorous physical activity and 31% in ‘time in bed’. This may indicate that the overall sample does relatively little physical activity. The representativeness of the sample could be discussed.
The large sample size allows the investigation of subgroups (moderator analysis). Did the results vary by menstruation cycle, hormone contraceptive use, etc.?
The practical implications could be illustrated in more detailed examples.
What do the findings indicate about achievements in the school context?
The consideration of BMI, obesity, or other indicators may be useful. Maybe this can be discussed in more depth.
The overall health status may be a consequence or cause of the individual physical activity.
At least these issues could benefit the discussion of findings.
Reviewer 2 Report
This research, based on chronotype status of 965 adolescent females, with measures of sleep and activity patterns, is scientifically sound, and also of clinical significance.
I would like clarification of the following points:
Were all the subjects post-menarche?
What hormonal correlates (such as progesterone levels) might be significant for the sleep and activity patterns of this group of females?
Since Leicester probably has the highest proportion of Asian-origin families in the UK, what proportion of Ss had this ethnic origin (assuming that it is relevant to ask about ethnicity, as the authors of this paper have done)?
Reviewer 3 Report
The theme of this project is very important for understanding what causes the increase in obesity among children and adolescents. In the last decades all over the world, the population tended to adopt a sedentary life and become overweight.
By talking about the chronotype you are shading light upon this subject of sedentarism and obesity. In my opinion, your paper should connect the chronotype with the type and number of meals for each participant in the study. I am not convinced that chronotype pers se is the key factor determining obesity, sedentarism, and psychiatric disorders among adolescents.
You should also check the time spent on social networks like Facebook, Instagram, Tik Tok or Twitter. Adolescents are captivated by social media and spend way too much time on social networks.
In conclusion, your study needs to be improved by verifying the role of food consumed by adolescents and the time spent on social networks, in determining body weight and behavior.
Reviewer 4 Report
The paper focuses on the sleeping patterns (chronotypes) among adolescent girls and their association with physical behaviors. The findings from the study reveal the prevalence of evening chorotype among the sample of adolescent females. Based on a review of the existing research, the authors underline that this result is associated with higher risks of poor mental and physical health and poor academic achievements that require special interventions. The paper addresses an important topic in the field of adolescent health and the study is well designed but in my opinion, the analysis needs to address the following issues.
At first, it is needed to explain why the study focuses only on girls and to include references on the gender differences in the sleeping patterns (chronotypes) of boys and girls. The results show that compared to the morning chorotypes, evening chronotypes engaged in more sedentary time and had lower overall physical activity at weekdays. The high prevalence of the evening chronotypes in the sample can influence the results of statistical analysis. In this regard, my question is if information was collected about girls’ academic performance, social activities (meeting with friends, outdoor activities/meetings, use of social networks and screening time) as well as specific types of health risk behaviors. Such information can describe better the activities that may have important consequence for (mental and phsical) health and the academic achievements of girls with different chorotypes. The organization of time for studies and for preparation of homework can also be in an association with prevalence of the different chronotypes. In case such information is collected in the study, it could be included in the paper in order to provide more information on the similarities and the differences in specific activities in which girls with different chronotypes were engaged. My suggestion to the authors is also to add a separate section about the limitations of the study and the implications for practice. It is also needed to extend the conclusion since in the present version of the paper it includes only two sentences that repeat the main results from the study.
Reviewer 5 Report
Dear Authors,
Congratulations on your extensive work Association between chronotype and physical behaviours in 2 adolescent girls
I suggest some minor revision:
Abstract:
The authors should start with some short theoretical background.
Introduction- is too concise.
M&M:
2.1. Sample
In my opinion, this paragraph should be divided into:
Study design, then inclusion and exclusion criteria, then recrutiment to the study, then procedures, then ethical issues.
In addition, how about adding a participant flowchart?
Discussion
I suggest adding separate paragraphes: strenghts and limitations of the current studies; future research inplications; practical clinical implications
Round 2
Reviewer 3 Report
The authors gave me a response for each of my issues.
I understand they have a different approach to the subject than the one I aimed at.
Given that the authors introduced more details about their study, now the paper is upgraded and the provided data better support it.
In these conditions, I consider that the paper should be published.
Author Response
We would like to thank the reviewer for their comments and feedback. We are glad that they are satisfied and accepted our responses.
Reviewer 4 Report
The suggested revisions are taken into consideration and the editing of the manuscript is satisfactory. This is why, I accept the present version. I would suggest editing of the graphs - color scheme, fonts, display.